# Predicting Physician Consultations for Low Back Pain Using Claims Data and Population-Based Cohort Data—An Interpretable Machine Learning Approach

**DOI:** 10.3390/ijerph182212013

**Published:** 2021-11-16

**Authors:** Adrian Richter, Julia Truthmann, Jean-François Chenot, Carsten Oliver Schmidt

**Affiliations:** 1Department SHIP-KEF, Institute for Community Medicine, Greifswald University Medical Center, Walther Rathenau Str. 48, 17475 Greifswald, Germany; carsten.schmidt@uni-greifswald.de; 2Department of Family Medicine, Institute for Community Medicine, Fleischmannstr. 42, 17475 Greifswald, Germany; julia.truthmann@med.uni-greifswald.de (J.T.); jchenot@uni-greifswald.de (J.-F.C.)

**Keywords:** record linkage, machine learning, calibration, best subset selection, low back pain

## Abstract

(1) Background: Predicting chronic low back pain (LBP) is of clinical and economic interest as LBP leads to disabilities and health service utilization. This study aims to build a competitive and interpretable prediction model; (2) Methods: We used clinical and claims data of 3837 participants of a population-based cohort study to predict future LBP consultations (ICD-10: M40.XX-M54.XX). Best subset selection (BSS) was applied in repeated random samples of training data (75% of data); scoring rules were used to identify the best subset of predictors. The rediction accuracy of BSS was compared to *randomforest* and *support vector machines* (SVM) in the validation data (25% of data); (3) Results: The best subset comprised 16 out of 32 predictors. Previous occurrence of LBP increased the odds for future LBP consultations (odds ratio (OR) 6.91 [5.05; 9.45]), while concomitant diseases reduced the odds (1 vs. 0, OR: 0.74 [0.57; 0.98], >1 vs. 0: 0.37 [0.21; 0.67]). The area-under-curve (AUC) of BSS was acceptable (0.78 [0.74; 0.82]) and comparable with *SVM* (0.78 [0.74; 0.82]) and *randomforest* (0.79 [0.75; 0.83]); (4) Conclusions: Regarding prediction accuracy, BSS has been considered competitive with established machine-learning approaches. Nonetheless, considerable misclassification is inherent and further refinements are required to improve predictions.

## 1. Introduction

Non-specific low back pain (LBP) is one of the leading health problems and associated with substantial individual burdens, health service utilization and indirect costs worldwide [1,2,3]. LBP has been portrayed as having a good prognosis with a high proportion of patients recovering spontaneously within few weeks, with no or minimal interventions. Although this is true for most, a substantial proportion of patients will develop chronic or recurrent back pain [4,5,6]. This has implications for the provision of health services.

One rationale is to focus service provision on subjects with subsequent high service utilization, reflecting relevant impairment and suffering. For health care providers and health insurances, the identification of such patients is, therefore, of great importance, since this is a prerequisite for the rational allocation of limited specialized services and for offering targeted interventions to those who are most affected. This would, for example, be of relevance to optimize disease management programs for chronic low back pain, which are mandated by the statutory health insurance in Germany.

Machine-learning approaches such as *randomforest* [7] have been shown to obtain high accuracy for the prediction of events and have frequently outperformed established models [8]. Nevertheless, the deficiencies of machine-learning approaches, including *randomforest,* comprise the following: (a) a black box like behavior, (b) lacking interpretability of results, and (c) the results cannot be reused in new data sources as the model definition is conducted implicitly [9,10]. 

In *best subset selection* (BSS) approaches, all possible combinations of candidate variables are explored in a regression model, i.e., 2^P^-1 possible subsets must be examined. This approach was proposed decades ago and provides accurate and interpretable prediction models, as candidate variables, which do not affect the prediction accuracy of the model, will be removed [11]. BSS is also known as L_0_—penalization and represents one among many penalized approaches, such as the *lasso* [12]. Due to its computational costs, BSS has rarely been applied in health research studies, although the availability of high-performance-cluster machines and parallel computing make BSS more feasible. 

In this study, we follow the pipeline of machine-learning approaches (model training, model calibration, and model validation) to obtain the best subset of predictors from claims data and those of a population-based cohort to predict future consultations for low back pain. For benchmarking, BSS results will be compared with parameter-tuned *randomforest* and *support vector machines*.

## 2. Materials and Methods

### 2.1. Data Sources

#### 2.1.1. Study of Health in Pomerania

The Study of Health in Pomerania (SHIP) [13] is a population-based project that comprises two distinct cohorts, SHIP-START and SHIP-TREND. This study is restricted to the baseline examinations conducted for the more recently started SHIP-TREND cohort (SHIP-TREND-0). Participants were sampled in Mecklenburg-Vorpommern, a region in northeast Germany, and followed up every 4–6 years. After the exclusion of deceased and relocated individuals, the net sample comprised 8826 individuals. In total, 4420 (2275 women) followed the invitation (response 50.1%) and were examined during the baseline visits between 2008 and 2012.

The SHIP-TREND-0 examination program comprised medical and dental examinations, laboratory measurements, an interview, and self-reported questionnaires, among others. In the latter, several standardized instruments were presented to participants, comprising the PHQ-9 [14], a diagnostic instrument for depressive symptoms, and a von Korff short version [15,16] to grade the severity of LBP.

#### 2.1.2. Claims Data

In Germany, Statutory Health Insurances host data of approximately 72.8 million assured individuals; 87.7% of the whole population [17]. For SHIP participants’, respective data were available between 2002 and 2018. We considered ICD-10 codes specific to diseases and symptoms in the lumbar spine (Dorsopathies, M40.XX to M54.XX), excluding ICD-10 codes of thoracic and cervical diagnoses (e.g., M54.01 to M54.04). Excluded ICD-10 codes are listed in the supplement under paragraph “List of ICD-10 codes” and Appendix A. In addition, physicians’ fee schedules were considered and grouped into those being claimed for back-pain-associated treatments by General practitioners, Orthopedists, Neurologists, Radiologists or for respective medical interventions. The list of included physicians’ fee schedules is provided in the supplement (Appendix A).

#### 2.1.3. Record Linkage

Data from the statutory health insurance and from SHIP have no common identifier for record linkage. Therefore, we followed the approach of Vatsalan et al. [18] and linked the data sources based on surname, name, date of birth, and sex of participants, with linkage consent. The names were normalized to upper-case letters; the indexing of candidate pairs was blocked via birth date. All exact agreements were considered matches and possible matches, i.e., agreements with Levenshtein distance > 0, were manually reviewed.

### 2.2. Methods

Data preparations and record linkage were conducted using SAS version 9.4 (SAS Institute Inc., Cary, NC, USA). R version 4.0.3 has been used for all analyses [19], including zero-inflated regression models using the R package *pscl* [20,21]. For the best subset selection, the R code was parallelized [22] and computed at the high-performance-computing center of the University of Greifswald [23].

#### 2.2.1. Study Design

For each SHIP-TREND-0 participant, a longitudinal analysis period covering three years of observation was considered (Figure 1). The year antedating the SHIP examination served as the baseline period, which also covers physicians’ fee schedules and ICD-10 codes from four quarters (Q_-3_ to Q_0_). This reflects the billing periods in ambulatory care. The year before the baseline period (Q_-7_ to Q_-4_) was used to identify any previous occurrences of ICD-10 codes related to unspecific low back pain. Claims data of quarters Q_1_ to Q_4_ represent follow-up data after participation in SHIP (Figure 1).

#### 2.2.2. Primary Outcome

The primary outcome was the sum of ICD-10 codes specific to diseases and symptoms in the lumbar spine during the year after baseline (Q_1_ to Q_4_, Figure 1). Assigned ICD-10 codes are a valid surrogate measure for physician consultations, as they can only be assigned following a consultation. This outcome represents count data, which were examined for zero-inflation using a score test [24,25] (Appendix A). 

#### 2.2.3. Candidate Variables

The following 12 candidate variables were considered for best subset selection from claims data: categorized age (<40; 40–69; > 69), history of LBP (yes/no prior to baseline), imaging procedure related to back pain in the baseline period, medical interventions related to back pain in the baseline period, sum of back-pain-related ICD-10 codes (in the quarter of the SHIP examination only), sum of back-pain-related ICD-10 codes (in at least two quarters of the baseline period), visited physician specialties in the baseline period (none, general practitioner only, specialist only, general practitioner and specialist), use of opioids, use of benzodiazepine, occurrence of ICD-10 code M54.XX only (unspecific back pain), the Charlson comorbidity index [26], and the interaction between categorized age and unspecific M54.XX only. (XX = any code after the decimal point is selected, except for those excluded and listed in the Supplement).

All 32 candidate variables from SHIP are listed in the supplement (Appendix A, Appendix A). This list was created based on the expert opinion of the investigators, and an informal literature review. Non-linear associations between candidate variables and the primary outcome were examined using generalized additive models (Appendix A). In addition, three interaction effects were assessed: depression x sex, depression x PHQ-9, and disc prolapse x radiating back pain. To reduce overall computational costs, we applied an exploratory variable selection using a boosting approach with stability selection [27] (please see supplement paragraph: “Exploratory variable selection”). This method has previously been applied to high-dimensional and collinear data [28]. In the case of strong non-linear associations (effective degrees of freedom > 1 and *p* < 0.05), the respective candidate variable was modelled as having a linear as well as a non-linear effect. With respect to the candidate variable age, the exploratory analyses favored a discretized form of age over a smoothed variant (B-spline) and a linear form. Therefore, discretized age was also used in the claims data. Overall, 19 candidate variables and one interaction effect were examined in the best-subset selection approach using the SHIP data. 

The best subsets of claims data and SHIP data were joined afterwards and assessed in another best subset selection approach in the joined data.

#### 2.2.4. Model Training

The whole model training is illustrated and annotated in the supplement (Appendix A). In brief, we first split the SHIP-TREND-0 data set into training data and validation data in a ratio of 3:1 (training data: n = 2869, validation data: n = 968). From the training data, b = 500 repeated bootstrap samples were drawn to define inner training data (observations selected by bootstrap) and inner validation data (observations not selected by bootstrap). Then, hurdle-models comprising a zero-part (logistic regression) and a count-part (either a *Poisson* or a negative-binomial distribution) were trained on the inner training data. The resulting model was used to predict the outcomes on the inner validation data. The model training process was iterated over two entities: (i) all possible combinations of covariates (*p_i_*) in bootstrap *b_j_*, and (ii) b = 500 bootstrap samples of the training data. This approach is similar to the one proposed by Filzmoser et al. for parameter tuning using repeated double cross-validation [29]. All these steps were separately conducted for the claims data and the SHIP data and, after the selection of the best subsets in each, the data were joined (Claims + SHIP).

#### 2.2.5. Model Evaluation

In each iterative step (*p_i_, b_j_*), information criteria (AIC, [30]) of *Poisson* and negative-binomial models were used to evaluate the best distributional form. In addition, strictly proper scoring rules were applied to assess the quality or agreement between the predicted and observed outcomes in the inner validation data [31]. The best model, i.e., the best subset of candidate variables, was considered to be the one that had, on average, the best prediction accuracy over 500 bootstrap samples. The selected model was then used to predict the outcomes in the outer validation data. 

The goodness of fit of the optimal model was examined using rootograms, which illustrate overfitting and underfitting of count data regression models [32]. Furthermore, the estimated probability of having an ICD-10 code (yes or no, which corresponds to the zero-part in the hurdle model) was used to predict and classify if study participants will have an ICD-10 code after participation in SHIP. The area-under-curve (AUC) and receiver-operating-characteristics (ROC) were generated for the best subsets of all three datasets [33] and the sensitivity analyses.

#### 2.2.6. Missing Data

Table 1 presents the proportion of missing values in all covariates originating from the SHIP study. First, we separately applied multiple imputations by chained equations (m = 20) using the R package mice [34] for training data and validation data. Second, the increase in variance in the training data (Appendix A) was evaluated for each candidate variable with missing values. The highest increase in variance was found for household income (4.58%), PHQ-9 (3.74%), and SF12 PCS sum score (1.24%). In all other candidate variables, the increase was <1%. Recommendations for the application of multiple imputations in the case of missing data < 5% are contradictory [35,36]. Therefore, for model training, we used similar training data in best subset selection and *randomforest* restricted to m = 1 imputation. However, to evaluate the best subset model in the validation data, we used m = 20 imputations.

#### 2.2.7. Comparative Analyses

To compare the predictive accuracy of the best subset model with established machine-learning algorithms we applied *randomforest* (RF) [7] and *support vector machines* (SVM) [37] using the combined data (claims and SHIP). As in most machine-learning approaches [38], *randomforest* and SVM have several tuning parameters to increase the predictive accuracy. For *randomforest*, we used the R package *tuneRanger* to define the optimal parameter setting [39]. For *SVMs,* we chose a radial basis function kernel due to non-linear associations and applied a two-step grid search for the best parameter setting [40]. The R package *e1071* was used to apply *SVMs* [41]. More details regarding the parameter tuning are provided in the supplement, under the paragraph “Comparative analyses”.

#### 2.2.8. Subgroup Analyses

Subgroup analyses were applied to examine sources for misclassification in predictive models. Therefore, estimated probabilities for a future episode of LBP were stratified in multiway-tables using the applied method, the training data and the validation data, back pain severity, and pattern of search of care.

## 3. Results

### 3.1. Cohort Characteristics

The descriptive characteristics of 4420 participants of the SHIP-TREND-0 cohort at baseline are summarized in Table 1. Of these, 341 (7.7%) were stated to be insured in a private or other health insurance organization and excluded from the analyses. Due to a temporarily reduced examination program, the insurance type was not asked for in 57 out of 4420 (1.3%) participants’; however, 51 of these consented to record linkage and were subsumed under the respective category. Therefore, out of the 4079 participants, 3837 (94.1%) consented to record linkage of claims data and were included in the analyses. Participant characteristics differed slightly between those consenting vs. not consenting record linkage. For example, mean age of consenting participants was 1.7 years higher compared to those not giving consent (Table 1). Larger differences, particularly regarding the household income, were found in participants insured in private/other health insurances (Table 1).

In 30% out of those consenting to record linkage, at least one ICD-10 code suggestive for LBP occurred within the baseline period. The distribution of the number of LBP-related ICD-10 codes was zero-inflated in both the baseline period and the follow-up period (Appendix A).

### 3.2. Best Subset Models

The best subset of candidate variables in the joined data comprised eight predictors for the zero-part and eight, partly different predictors for the count-part of the hurdle model (Table 2). According to AIC, a Poisson distribution of the primary outcome was preferred over a negative-binomial distribution in all models and iterations of the model training. Overall, the fit of a hurdle model using the best subset of predictors with a Poisson distribution for the count-part was high in the training and validation data (please see rootograms [32] in Appendix A).

In the zero-part of the hurdle model using the claims and SHIP data, participants between 40 and 69 years of age had higher odds of future consultations compared to the reference group of individuals under 40 years of age, while participants older than 69 years had considerably lower odds (Table 2). The odds for consultations increased with a higher household income, a history of disc prolapse, and female sex. In contrast, the presence of chronic diseases (cardiovascular disease, cancer, diabetes) lowered the odds for future consultations. The strongest association was observed for the number of ICD-10 codes in the baseline period and a history of back pain (prior baseline) (Table 2). If ICD-10 codes suggestive for LBP were found in at least two quarters of the baseline period (2Q), the odds for a future event of LBP increased (odds ratio (OR): 2.43, 95% confidence interval (CI) [1.97; 2.99]), whereas, in contrast, if ICD-10 codes were found only in the quarter of the SHIP examination (acute) the odds decreased (OR: 0.44 [0.35; 0.56]).

For the count-part of the hurdle model, different candidate variables were selected. The count of back-pain-related ICD codes increased with the number of ICD-10 codes (2Q) documented during the baseline period. In addition, higher back pain (numerical rating scale (NRS), radiating back pain, and rare physical activity (1–2 h/week vs. none and >2 h/week) increased the count. The use of benzodiazepines and being single (compared to married/widowed/separated) lowered the counts (Table 2). Participants’ sex had no effect on the count.

The predictive value of most candidate variables from SHIP data were lower compared to candidate variables from the claims data (see Brier scores and Dawid–Sebastiani scores in the Supplement). For example, self-reported physician visits was a strong predictor within the best subset of the SHIP data. In the joined data, however, this predictor was not selected. In contrast, competing diseases, which were defined based on SHIP data, remained in the best subset of the joined data, as well as participant-reported outcomes of back pain severity (NRS), radiating back pain, and household income.

### 3.3. Predictive Accuracy and Comparative Analyses

The best subset model using only SHIP data performed lowest in predicting future LBP-related consultations with an area-under-curve (AUC) of 0.68 (95% CI [0.64, 0.72]) (Figure 2). This is considered a non-acceptable discrimination [42]. The accuracy of the best subset model using claims data (AUC: 0.76 [0.72; 0.80]) was higher, and barely improved on by the best subset of the joined data (AUC: 0.78 [0.74; 0.82]). The presence of missing data had no influence on the predictive accuracy of the best subset model. The receiver-operating-characteristic’s (ROCs) for the different imputations of the validation data were almost identical (Appendix A).

For comparative analyses using randomforest, the optimal parameter setting was estimated using tuneRanger [39], which resulted in: mtry = 9 (number of variables used for splitting the trees), ntree = 10,000 (number of trees), nodesize = 60 (minimum terminal node size), sampsize = 1674 (sample size of learning data)). The tuned randomforest model obtained, by a slight margin, the highest accuracy (AUC: 0.79 [0.75; 0.83]). Parameter-tuning of SVM resulted in a constant C of 1.3 and a γ–value of 0.02. The result of SVMs was similar to BSS (AUC: 0.78 [0.74; 0.82]).

According to a variable importance measure of *randomforest*, the number of ICD-10 codes documented in the baseline period, age, and a history of back pain (prior baseline) had the strongest importance for the prediction of future episodes of LBP (Appendix A).

### 3.4. Subgroup Analyses

Study participants with incident consultations of LBP in the follow-up period only, i.e., those with no history of back pain and no ICD-10 codes suggestive for LBP during the baseline period, had, according to both model approaches, low probabilities for future LBP related consultations. The estimated probabilities were very similar for those with no ICD-codes suggestive for LBP (Figure 3). As shown in the Appendix A, these two strata of study participants share very similar characteristics, except for participants’ sex (seek of care: 0|0 = 48.8% females, 0|1= 58.1% females, Χ^2^ = 9.21, *p* < 0.01) and back pain in the last 3 months (µ_diff_ = 0.49 scores on NRS, *p* < 0.01 (*t*-test)). Mean age was almost identical between these two strata. However, participants with health care utilization only after baseline (0|1) were more frequently aged from 40 to 70 years (Χ^2^ = 22.55, *p* < 0.01), a subgroup that has been found to be more susceptible to seek care for LBP (Table 2).

In addition, participants with a history of back pain and/or ICD-codes suggestive for LBP in the baseline period but no ICD-codes in the follow-up periods had comparable probabilities to those seeking continuously physicians care, i.e., having ICD-10 codes suggestive for LBP in all three analysis periods. The former participants were slightly older than those steadily seeking care (µ_diff_ = 2.4 years, *p* < 0.01 (*t*-test)), and more frequently competing diseases (OR: 1.58 [1.23; 2.03]), and lower household income (median: 1096 vs. 1364, Wilcoxon rank sum test: *p* < 0.01). The abovementioned patterns were similar across both methodological approaches, the training and validation data, and irrespective of back pain severity (Figure 3).

## 4. Discussion

We applied best subset selection (BSS) in a common machine-learning pipeline of model training in resampled data, model calibration and model averaging using strictly proper scoring rules, as well as internal model validation of predictions of future consultations for low back pain (LBP) in the general population. The predictive accuracy of BSS (AUC: 0.78) was comparable to the established machine-learning approaches *randomforest* (*SVM*) (AUC: 0.79 (0.78)). The accuracy of all approaches is acceptable [42] and in line with those reported for other machine-learning-based prediction models [8,43] and for different back-pain-related outcomes [44]. In terms of computational time and efficacy, BSS (21 h) was outperformed by *randomforest* (4:01 min) and SVM (24:46 min). From a clinical perspective, the benefit of interpretable model coefficients, as provided by BSS (Table 2), outweighs this computational drawback.

An acceptable accuracy (AUC in [0.7; 0.8], [42]), as in our study, is commonly reported for prediction models of health research studies [43]. Regarding the prediction of LBP, however, most previous studies lacked the validation of predictors as summarized by a systematic review [45]. Therefore, in this clinical domain, our study represents one among a few validated and clinically interpretable prediction models.

The best subset of predictors in the joined data comprised strong predictors from claims data, particularly variables of previous consultations for LBP, and participant-reported outcomes collected in the SHIP study, such as back pain severity (NRS), radiating back pain, and household income. The latter outcomes were inferior in their predictive value compared to the predictors from claims data (Appendix A). Therefore, the AUC of the best subset in the joined data improved only marginally compared to the best subset using claims data only (Figure 2). The observed lack of improvement in predictions with participant-reported outcomes was also indicated by the low-to-moderate univariate associations of self-reported back pain outcomes and the frequency of LBP-related ICD-10 codes (Appendix A). Therefore, it seems that past administrative data are best at predicting future administrative data.

From a clinical perspective, this rather surprising finding reflects the fact that claims data in this study represent data from patients who decided to seek medical care for LBP. The clinical data from the SHIP study reflect the data of the general population. The results of our study suggest that many participants with LBP were able to cope with LBP without seeking medical care, and thus did not become patients. As was shown in other research, health-care-seeking behavior is partially detached from clinical symptoms [46]. Another important factor explaining differences in the predictive value of clinical data and claims data is comorbidity or competing diseases. Participants with higher Charlson comorbidity index (CCI) or competing disease were less likely to reconsult for LBP (Table 2). This does not reflect lower pain severity, but a higher likelihood of seeking and obtaining care for a predominant health problem and a lower likelihood of LBP being coded in the claims data. This role of coping strategies and competing diseases was also affirmed by the systematic review by Ferreira et al. [47]. 

In subgroup analyses of our study, we found that a considerable proportion of participants (n = 740, 19.3%) with severe LBP had previously consulted a physician for LBP, but these participants did not seek physicians’ care again during follow-up. The respective participants were of higher age, more frequently retired, had more competing diseases, and a lower household income. The characteristics of this subgroup had strong implications on the predictive accuracy of all methodological approaches in this study. These participants previously sought care for LBP, had a high intensity of back pain, and in 46.9%, back pain was radiating to lower extremities (Appendix A). In consequence, very high probabilities for future episodes of LBP were estimated from all prediction approaches (Figure 3). Irrespective of any cutoff used to balance sensitivity and specificity in classification [48], this subgroup of participants will be misclassified by present approaches (Figure 3). 

In this study, claims data covering only two years were included in the prediction model (Figure 1). Extending the pre-observation period might have allowed for the identification of patients who used extensive medical services for LBP in the past and, therefore, discontinued seeking care despite persisting pain, e.g., due to unsuccessful treatment attempts or exhaustion of treatment options. Identifying this longitudinal pattern in seeking care for LBP might improve prediction models, possibly under consideration of competing diseases such as cancer.

A second subgroup susceptible to misclassification were participants with incident LBP episodes in the follow-up only (Figure 3). Except for participants’ sex and a slightly higher back pain intensity, these participants were almost similar compared to those not seeking care for LBP (Appendix A). Recently, a similar difficulty was found in the correct prediction of incident LBP in a large claims data-based study for the new onset and recurrence of LBP with over 400,000 participants [49]. 

Strength and Limitations: Our analyses were based on a large general population sample, with a very high coverage of claims data. While we considered a wide ange of predictors from SHIP and claims data, LBP and potential risk-factors could have been characterized in more depth to further enrich the power of potential predictors, e.g., by more widely covering psychosocial predictors, for example, fear-avoidance beliefs [44,50]. In addition, further refinements in the study design must be sought to more appropriately select persons at risk of unfavorable LBP courses at different clinical states.

The best subset selection, as applied in our approach, is not a true machine-learning approach. Although we emphasize that there is no clear distinction between statistical modeling and machine-learning [43], advanced regression techniques such as *lasso*—which learns from the data and requires parameter tuning—are sometimes not considered machine-learning [43], whereas the application of logistic regression for prediction is sometimes, debatably, considered a machine-learning approach [8]. We consider our approach an interpretable or hybrid form, which mimics the pipeline of machine learning while maintaining interpretability: the approach was conducted to make robust predictions. From the data, we learned the optimal combination of candidate variables and the best fitting distribution, and we gained insight into the clinical and socio-demographic roles of candidate variables.

We applied common statistical and epidemiological reasoning to reduce bias. For example, unlike many machine-learning applications, in which either single-value imputations are used [8] or missingness in data elements is not mentioned at all [51], we explicitly addressed the uncertainty of imputations using multiple imputations in the validation data, and independently of the training data. The best subset model was evaluated in the validation data for 20 imputed versions of the data (Appendix A). In this study, the impact of missing data was negligible.

In *randomforest,* multiple interactions between candidate variables are implicitly accounted for, which can be investigated, for example, using the R package *randomForestExplainer* [52]. This feature of *randomforest* might explain the slightly better performance compared to best subset selection. The superiority of *randomforest,* as well as *SVM*, was found for computational efficacy. This computational drawback of BSS will be mitigated as soon as advances in mixed-integer-optimization (MIO) are extended to use cases with data elements of mixed data types (continuous and categorical). Bertsimas et al. [53] have shown the possibility of solving more complex settings of best subset selection in minutes. 

The strongest limitation of our study is that we were not able to validate the model with external data. Our approach comprised internal validation, as all data were generated and collected under the same design. 

## 5. Conclusions

Claims data alone could be used to predict patients with LBP who are likely to reconsult for LBP with acceptable accuracy. Adding participant-reported outcomes from a population-based cohort study to the prediction model yielded a minor improvement, but only in the current unstratified setting. Therefore, further refinements of patients at risk are required to obtain insights into the role of predictors at different patient states over the course of LBP. 

The development of prediction models in health research should comprise epidemiological and statistical reasoning to minimize bias, maintain the interpretability of results, and pass through a pipeline of model training, calibration, and validation. These efforts can result in prediction models, with a comparable performance to machine-learning approaches. The latter, such as *randomforest,* should be applied more frequently, to benchmark and possibly improve the models proposed by other modelling techniques in case of a major inferiority. Overall, and given a comparable performance, the interpretability of the study results outweighs computational efficacy in health research. In this manner, best subset selection appears a reasonable modeling approach.

## Figures and Tables

**Figure 1 ijerph-18-12013-f001:**
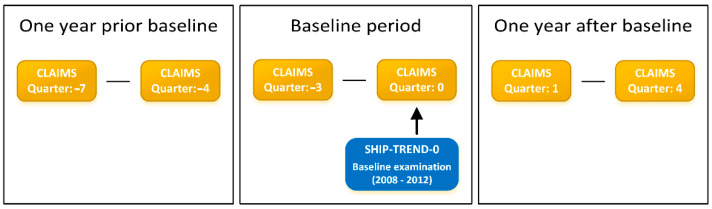
Study design overview.

**Figure 2 ijerph-18-12013-f002:**
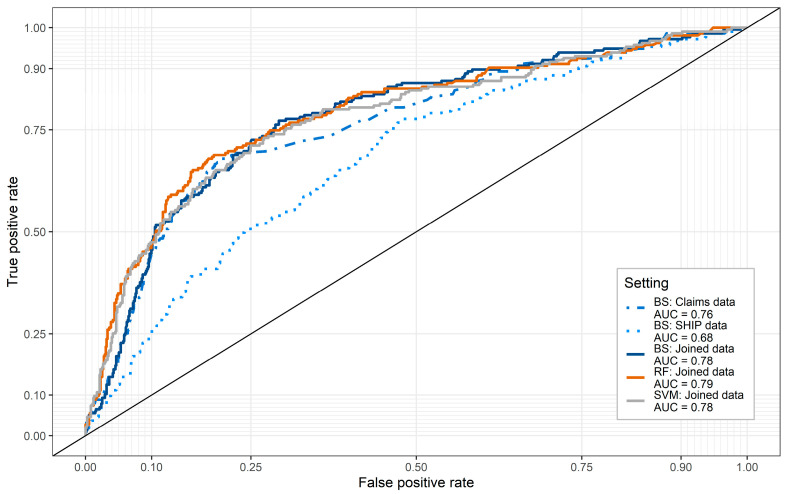
Area under curve (AUC) and receiver-operating-characteristics (ROC) for different best subset models, *randomforest, and support vector machines* in predicting future LBP-related consultations of physicians. (BS = best subset, RF = randomforest, SVM = support vector machines).

**Figure 3 ijerph-18-12013-f003:**
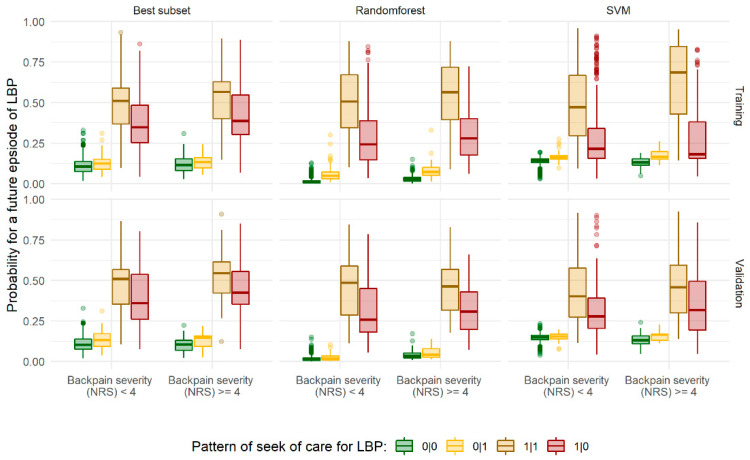
Predicted probabilities of future consultations for low back pain (LBP) stratified by the applied method (best subset vs. *randomforest* and *support vector machines (SVM)*), backpain severity (NRS), data source, and consultation patterns of seeking care. (Seek of care: 0|0 = no ICD-10 codes suggestive for LBP in any of the analysis periods (N = 2229), 0|1 = ICD-10 codes suggestive for LBP only during follow-up (N = 267), 1|1 = ICD-10 codes suggestive for LBP in all analysis periods (N = 601), 1|0 = history of back pain prior baseline and/or ICD-10 codes suggestive for LBP at baseline but not in the follow-up (N = 740)).

**Table 1 ijerph-18-12013-t001:** Characteristics of participants of the Study of Health in Pomerania TREND-0 cohort.

Characteristic	Statutory Insurances		Private Insurances
	Participants with Consent to Linkage	Participants without Consent to Linkage	
N	3837	242	341
Age (years)			
Mean (SD)	52.6 (15.6)	50.9 (15.3)	45.2 (12.1)
Median [Min, Max]	53.0 [20.0, 84.0]	52.5 [23.0, 81.0]	44.0 [21.0, 78.0]
Family status			
Single	409 (10.7%)	40 (16.5%)	36 (10.6%)
Married/Partner	2964 (77.2%)	170 (70.2%)	284 (83.3%)
Separated	241 (6.3%)	24 (9.9%)	18 (5.3%)
Widowed	211 (5.5%)	7 (2.9%)	3 (0.9%)
Missing	12 (0.3%)	1 (0.4%)	0 (0%)
Job characteristics			
Never working	27 (0.7%)	2 (0.8%)	5 (1.5%)
At desktop, not physically	1129 (29.4%)	75 (31.0%)	188 (55.1%)
At desktop and physically demanding	504 (13.1%)	36 (14.9%)	64 (18.8%)
Not at desktop, not physically	796 (20.7%)	49 (20.2%)	28 (8.2%)
Not at desktop but physically demanding	1220 (31.8%)	61 (25.2%)	52 (15.2%)
Missing	161 (4.2%)	19 (7.9%)	4 (1.2%)
School years			
<10	954 (24.9%)	64 (26.4%)	11 (3.2%)
10	2018 (52.6%)	104 (43.0%)	146 (42.8%)
>10	853 (22.2%)	73 (30.2%)	184 (54.0%)
Missing	12 (0.3%)	1 (0.4%)	0 (0%)
Household income			
Mean (SD)	1300 (638)	1290 (704)	2140 (913)
Median [Min, Max]	1100 [149, 5070]	1100 [192, 3580]	2050 [149, 5070]
Missing	134 (3.5%)	33 (13.6%)	20 (5.9%)
Backpain last 3M (NRS)			
Mean (SD)	2.69 (2.63)	2.48 (2.57)	1.89 (2.24)
Median [Min, Max]	3.00 [0, 10.0]	2.00 [0, 10.0]	1.00 [0, 10.0]
Missing	14 (0.4%)	1 (0.4%)	0 (0%)
Impairment by backpain last 3M (NRS)			
Mean (SD)	1.02 (1.89)	0.996 (1.98)	0.619 (1.53)
Median [Min, Max]	0 [0, 10.0]	0 [0, 10.0]	0 [0, 10.0]
Missing	15 (0.4%)	1 (0.4%)	0 (0%)
PHQ-9			
no signs	513 (13.4%)	36 (14.9%)	57 (16.7%)
minimal	1907 (49.7%)	95 (39.3%)	174 (51.0%)
mild	976 (25.4%)	59 (24.4%)	89 (26.1%)
moderate	204 (5.3%)	20 (8.3%)	17 (5.0%)
severe	61 (1.6%)	4 (1.7%)	0 (0%)
Missing	176 (4.6%)	28 (11.6%)	4 (1.2%)
Use of NSAIDs			
NSAIDs	383 (10.0%)	28 (11.6%)	27 (7.9%)
No NSAIDs	3444 (89.8%)	214 (88.4%)	314 (92.1%)
Missing	10 (0.3%)	0 (0%)	0 (0%)
Use of opioids			
Opioids	64 (1.7%)	10 (4.1%)	0 (0%)
No opioids	3763 (98.1%)	232 (95.9%)	341 (100%)
Missing	10 (0.3%)	0 (0%)	0 (0%)
Use of antidepressants			
No antidepressants	3607 (94.0%)	232 (95.9%)	331 (97.1%)
antidepressants	220 (5.7%)	10 (4.1%)	10 (2.9%)
Missing	10 (0.3%)	0 (0%)	0 (0%)
ICD-10 codes for backpain			
No	2684 (70.0%)	0 (0%)	289 (84.8%)
One	650 (16.9%)	0 (0%)	1 (0.3%)
Two	292 (7.6%)	0 (0%)	2 (0.6%)
≥3	211 (5.5%)	0 (0%)	1 (0.3%)
Missing	0 (0%)	242 (100%)	48 (14.1%)

**Table 2 ijerph-18-12013-t002:** Results of best subset selection for the claims data, the SHIP data, and the joined data to predict the number of LBP-related ICD-codes during follow-up.

Characteristics		Claims Data		SHIP Data		Joined Data	
	Model-Part	OR/IRR	95% CI	OR/IRR	95% CI	OR/IRR	95% CI
Age discrete (ref: <40 years)							
Age discrete (40 to 69 years)	Zero	1.80	[1.38; 2.35]	2.02	[1.56; 2.61]	1.73	[1.33; 2.27]
Age discrete (>69 years)	Zero	0.74	[0.51; 1.09]	1.10	[0.77; 1.56]	0.74	[0.51; 1.08]
Females (ref: males)	Zero			1.34	[1.11; 1.62]	1.34	[1.10; 1.64]
Household income (per 100 €)	Zero			1.03	[1.02; 1.05]	1.03	[1.02; 1.05]
Backpain intensity in last 3 month (NRS)	Zero			1.05	[1.01; 1.10]		
Radiating back pain (ref: none)							
gluteal only	Zero			1.47	[1.12; 1.93]		
to the knee	Zero			1.59	[1.15; 2.18]		
into lower leg	Zero			1.57	[1.07; 2.31]		
History of disc prolapse (yes vs. no) 1	Zero			1.69	[1.25; 2.29]	1.32	[0.95; 1.82]
SHIP physician visit (ref: none) 2							
General practitioner only	Zero			1.54	[1.15; 2.05]		
Specialist only	Zero			2.73	[1.62; 4.58]		
General practitioner and specialist	Zero			2.97	[2.16; 4.09]		
Competing diseases (ref: no)							
one	Zero			0.86	[0.66; 1.11]	0.74	[0.57; 0.98]
>one	Zero			0.46	[0.27; 0.80]	0.37	[0.21; 0.67]
Charlson comorbidity index	Zero	0.92	[0.87; 0.97]				
# ICD-10 codes related to lumbar spine (2Q) 3	Zero	2.50	[2.03; 3.07]			2.43	[1.97; 2.99]
# ICD-10 codes related to lumbar spine (acute) 3	Zero	0.45	[0.36; 0.58]			0.44	[0.35; 0.56]
History of backpain (yes vs. no) 4	Zero	6.75	[4.95; 9.21]			6.91	[5.05; 9.45]
Age (years)	Count			1.01	[1.00; 1.02]		
Females (ref: males)	Count	1.04	[0.88; 1.24]			1.00	[0.84; 1.20]
Family status (ref: single)							
Married/Partner	Count			1.57	[0.96; 2.57]	1.44	[0.88; 2.34]
Separated	Count			1.63	[0.90; 2.96]	1.27	[0.71; 2.29]
Widowed	Count			1.97	[1.09; 3.57]	1.60	[0.89; 2.87]
Work status (ref: employed)							
Retired	Count			0.97	[0.74; 1.26]		
Unemployed	Count			0.70	[0.48; 1.00]		
Physical activity (ref: none)							
1–2 h/week	Count			1.26	[0.98; 1.60]	1.19	[0.93; 1.52]
>2 h/week	Count			1.06	[0.79; 1.41]	1.04	[0.78; 1.39]
Backpain intensity in last 3 month (NRS)	Count			1.05	[1.01; 1.09]	1.01	[0.97; 1.05]
Radiating back pain (ref: no)							
gluteal only	Count			1.41	[1.11; 1.80]	1.20	[0.94; 1.54]
to knee	Count			1.54	[1.20; 1.99]	1.44	[1.12; 1.85]
to lower leg	Count			1.59	[1.18; 2.13]	1.34	[1.00; 1.78]
History of disc prolapse (yes vs. no) 1	Count			1.26	[1.02; 1.56]		
Osteoarthritis (yes vs. no)	Count			1.11	[0.92; 1.34]	1.07	[0.90; 1.27]
SHIP physician visit (ref: none) 2							
General practitioner only	Count			1.09	[0.77; 1.53]		
Specialist only	Count			1.24	[0.78; 1.97]		
General practitioner and specialist	Count			1.40	[0.99; 1.98]		
Use of benzodiazepine (yes vs. no)	Count	0.82	[0.43; 1.57]			0.79	[0.41; 1.52]
History of backpain (yes vs. no)	Count	1.35	[1.02; 1.79]				
# ICD-10 codes related to lumbar spine (2Q) 3	Count	1.53	[1.41; 1.66]			1.56	[1.45; 1.68]
M54.XX only	Count	0.49	[0.34; 0.70]				
Claims: Physician visit (ref: none) 5							
General practitioner only	Count	0.98	[0.72; 1.35]				
Specialist only	Count	0.88	[0.52; 1.48]				
General practitioner and specialist	Count	0.89	[0.64; 1.24]				

The table consists of two parts: the zero-part of the zero-inflated count data model, in which exponentiated coefficients correspond to odds ratios, and the count part, in which exponentiated coefficients can be interpreted as incidence rate ratios.^1^ SHIP interview item: disc prolapse ever before SHIP, ^2^ SHIP interview item: physician visits within last 4 weeks, ^3^ No. of ICD-10 codes (2Q: codes must have been reported in two quarters of the baseline period of 12 month; acute: any of the code appeared within the quarter of the SHIP examination), ^4^ Claims data: any ICD-10 code suggestive for LBP prior baseline (yes vs. no), ^5^ Claims data: based on physicians’ fee schedules. # = the quantity of respective ICD-10 codes.

## Data Availability

Data of the SHIP studies are available and can be applied for under https://www.fvcm.med.uni-greifswald.de/ (last accessed on 15 November 2021). The claims data utilized in this study are not publicly available due to privacy restrictions for the protection of personal data of research participants.

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
