# Peer review of "Predicting Physician Consultations for Low Back Pain Using Claims Data and Population-Based Cohort Data—An Interpretable Machine Learning Approach"

_ijerph, 2021, doi:10.3390/ijerph182212013_

Round 1

Reviewer 1 Report

This study employed machine-learning approaches to obtain the best subset of predictors from claims data and those of a population-based cohort to predict future consultations for lower back pain. For benchmarking, best subset selection results were compared with a tuned randomforest prediction model. In general, the manuscript is well written and provides the novel respective on the traditional health issue, that is, the LBP prediction. In the past, the multiple logistic regression method is the most popular tool to clarify the risk factors causing LBP. This paper may stand on the shoulders of giants to extend the application for the LBP prevention. This is really good. However, I still have some concerns about the manuscript before it can be accepted for publication in the journal.

  1. The study objective should be highlighted in Abstract. However, when reading Introduction, Results, Discussion, and Conclusion sections, the study objective seems to be a little inconsistent.
  2. When reporting “The best subset approach resulted in a prediction model with interpretable effect estimates and of comparable performance with randomforest. Computational costs were high but feasible.”, what is the primary contribution of this study? Can the proposed method improve the accuracy? increase the ability of variance explained? shorten the analyzing time? find out the results that other approaches fail to? or anything else? This should be clearly clarified.
  3. Discussion section is too long to be focused on. I suggest that the authors remove the irrelative contents and just show what and how they found from the analyses and then compare with that from the previous studies, or any implication for the future investigations.
  4. Several terms regarding machine-learning (ex. BSS, AUC…) should be spelled out when they appeared first time and then the abbreviations can be used to enhance the manuscript cleanness and readability.
  5. L45-47. When saying “Nevertheless, deficiencies of machine-learning approaches including randomforest comprise: (a) a black box like behavior, (b) lacking interpretability of results, and (c) the results cannot be reused in new data sources as the model definition is conducted implicitly [9,10].”, the proposed BSS seemed to cover these deficiencies and should be mentioned in the Discussion section.
  6. L481, The Supplementary Materials linkages are unavailable.

Author Response

We thank the Reviewer for these valuable comments. We have revised the manuscript thoroughly and provide detailed responses as well as the applied changes below.

[Issue 1] The study objective should be highlighted in Abstract. However, when reading Introduction, Results, Discussion, and Conclusion sections, the study objective seems to be a little inconsistent.

Response: We have revised the abstract and the manuscript to highlight the objective.

[Issue 2] When reporting “The best subset approach resulted in a prediction model with interpretable effect estimates and of comparable performance with randomforest. Computational costs were high but feasible.”, what is the primary contribution of this study? Can the proposed method improve the accuracy? increase the ability of variance explained? shorten the analyzing time? find out the results that other approaches fail to? or anything else? This should be clearly clarified.

Response: We submitted the manuscript on call for submissions under the issue “The State of the Art of Health Data Science”; manuscripts were invited if focusing – among others – on prediction models. In our perspective, the application of machine learning is frequently considered “State of the art” for prediction models. However, they lack interpretability which imposes ethical uncertainty in health research and medicine (Yoon et al. "Machine learning in medicine: should the pursuit of enhanced interpretability be abandoned?." Journal of Medical Ethics (2021). http://dx.doi.org/10.1136/medethics-2020-107102). Our contribution shows, that machine learning not necessarily provides better prediction models and thorough statistical modeling may be competitive. The latter is still not self-evident, particularly in this clinical domain (McIntosh et al., doi:10.1097/AJP.0000000000000591). Best subset selection (BSS), however, does not increase accuracy, or reduces computational costs, the advantage of BSS rests with competitive predictive accuracy and interpretability. We have emphasized the peculiarities of BSS in the discussion (end of paragraph 1).

[Issue 3] Discussion section is too long to be focused on. I suggest that the authors remove the irrelative contents and just show what and how they found from the analyses and then compare with that from the previous studies, or any implication for the future investigations.

Response: We removed several paragraphs of the discussion thereby reducing the word count of the discussion from 1,913 to 1,420.

[Issue 4] Several terms regarding machine-learning (ex. BSS, AUC…) should be spelled out when they appeared first time and then the abbreviations can be used to enhance the manuscript cleanness and readability.

Response: We have introduced all abbreviations.

[Issue 5] L45-47. When saying “Nevertheless, deficiencies of machine-learning approaches including randomforest comprise: (a) a black box like behavior, (b) lacking interpretability of results, and (c) the results cannot be reused in new data sources as the model definition is conducted implicitly [9,10].”, the proposed BSS seemed to cover these deficiencies and should be mentioned in the Discussion section.

Response: It seems like the Reviewer could not access the supplementary file (issue 6). This file comprises all steps of testing for the appropriate distribution, definition and selection of candidate variables, reports on missing data and many more (overall 28 pages). All these steps were applied to choose the appropriate modeling approach. In addition, in the discussion (version of revision + trackchange: p15 lines 451 - 456) we mention some of the benefits of our approach using BSS. In this regard we think, that the current version of the manuscript is comprehensive.

[Issue 6] L481, The Supplementary Materials linkages are unavailable.

Response: We are sorry for this inconvenience. We have uploaded the supplementary file during the submission process. It has also been made available open access (Zenodo) under the following doi: 10.5281/zenodo.5515295. Due to recommendations of Reviewer 2 we extended the scope of approaches and created a slightly changed version of the supplement (doi: 10.5281/zenodo.5590404). We hope that you find everything accessible.

Reviewer 2 Report

Both 'best subset selection' and 'random forest' approaches are existing approaches.
There exists other popular approaches, such as (1) SVM with various kernels and (2) feature selection with one of the existing classifiers.
Comparing only two approaches seems not enough. please add at least one more set of results using a third classifier.

Author Response

We thank the Reviewer for these valuable comments. We have revised the manuscript thoroughly and provide detailed responses as well as the applied changes below.

Both 'best subset selection' and 'random forest' approaches are existing approaches. There exists other popular approaches, such as (1) SVM with various kernels and (2) feature selection with one of the existing classifiers.

[Issue 1] Comparing only two approaches seems not enough. please add at least one more set of results using a third classifier.

Response: Thank you for this recommendation. We added the application of a parameter-tuned support vector machine to the manuscript. Please see the abstract, paragraph “Comparative analysis”, results, and the discussion. Methods and results of parameter tuning have also been added into the revised supplement (doi: 10.5281/zenodo.5590404). We omitted additional feature selection, e.g. using RFE, as there are contradictory results on the benefit of this approach; e.g. Chen et al. found SVM with highest accuracy without prior feature selection (Chen, Rung-Ching, et al. "Selecting critical features for data classification based on machine learning methods." Journal of Big Data 7 (2020): 1-26.)